# Psychological Impact of Newborn Screening for 3-Methylcrotonyl-CoA Carboxylase Deficiency: The Parental Experience

**DOI:** 10.3390/ijns11040115

**Published:** 2025-12-14

**Authors:** Vincenza Gragnaniello, Giacomo Gaiga, Chiara Cazzorla, Elena Porcù, Daniela Gueraldi, Andrea Puma, Christian Loro, Mara Doimo, Leonardo Salviati, Alberto B. Burlina

**Affiliations:** 1Division of Inherited Metabolic Diseases, Department of Women’s and Children’s Health, University Hospital of Padua, 35128 Padua, Italy; giacomo.gaiga@aopd.veneto.it (G.G.); chiara.cazzorla@aopd.veneto.it (C.C.); elena.porcu@aopd.veneto.it (E.P.); daniela.gueraldi@aopd.veneto.it (D.G.); andrea.puma@aopd.veneto.it (A.P.); christian.loro@aopd.veneto.it (C.L.); alberto.burlina@unipd.it (A.B.B.); 2Clinical Genetics Unit, Department of Women’s and Children’s Health, University of Padua, 35128 Padua, Italy; mara.doimo@unipd.it (M.D.); leonardo.salviati@unipd.it (L.S.)

**Keywords:** 3-methylcrotonyl-CoA carboxylase deficiency, newborn screening, tandem mass spectrometry, long-term follow-up, clinical outcome, metabolic decompensation, communication, emotional support, parental anxiety, psychological adjustment

## Abstract

3-Methylcrotonyl-CoA carboxylase deficiency (3-MCCD) is a metabolic disorder with a wide clinical spectrum ranging from asymptomatic individuals to severe metabolic decompensation. Following the introduction of expanded newborn screening, a high number of asymptomatic individuals with 3-MCCD were identified, prompting debates about its inclusion in screening panels. In order to inform policy and healthcare decisions regarding the inclusion of 3-MCCD in newborn screening programs, we evaluated the long-term outcomes for newborns with positive results over a decade of screening experience in North-East Italy, as well as the psychological impact on their parents. Of the 336,668 newborns screened between 2014 and 2025, 9 were confirmed to be affected. These infants underwent annual clinical and biochemical assessments, including dried blood spot acylcarnitine profile, plasma free carnitine, and urinary organic acids assays. An emergency protocol was provided to all affected children to manage intercurrent illnesses. An ad hoc survey was developed to assess the psychological impact of the disease on parents. During follow-up (mean age at last visit: 4.2 years), one patient experienced metabolic decompensation during an intercurrent illness, which was promptly treated. One patient presented with growth retardation and another with transient psychomotor delay. Five patients developed carnitine deficiency, requiring supplementation. Psychological assessments revealed an initial high level of parental psychological impact, which decreased over time. All parents strongly supported the screening program. Newborn screening for 3-MCCD enabled the early identification and management of affected individuals, thereby avoiding severe metabolic decompensation. Although there is an initial psychological burden on parents, it significantly decreases over time. Therefore, the long-term benefits of newborn screening for 3-MCCD seem to outweigh the psychological drawbacks.

## 1. Introduction

3-Methylcrotonyl-CoA carboxylase deficiency (3-MCCD, OMIM #210210) is an autosomal recessive metabolic disorder that affects leucine catabolism (see Figure 1). It results from impaired function of the mitochondrial enzyme 3-methylcrotonyl-CoA carboxylase (3-MCC), comprising two subunits (MCCα and MCCβ) encoded by the *MCCC1* and *MCCC2* genes, respectively [1,2].

Enzymatic deficiency leads to the accumulation of 3-methylcrotonyl-CoA within mitochondria, where alternative pathways form 3-methylcrotonylglycine (3-MCG) and 3-hydroxyisovaleric acid (3-HIVA). Binding to carnitine forms 3-hydroxyisovalerylcarnitine (C5OH) [3].

The clinical presentation of 3-MCCD is highly variable. Early reports described patients experiencing metabolic crises, including hypoglycemia with or without acidosis, ketosis and fasting intolerance. Children with the condition were also thought to be at risk of feeding difficulties, growth retardation, hypotonia, seizures and varying degrees of developmental and psychomotor delay [4,5].

Following the introduction of acylcarnitine analysis of dried blood spots (DBS) in newborn screening (NBS), infants with 3-MCCD can be identified through elevated levels of C5OH.

In NBS programs in North America, Europe and Australia, 3-MCCD is the most frequently detected organic aciduria [6,7,8,9,10]. Its prevalence ranges from 1 in 2400 to 1 in 68,000 [11], which is significantly higher than predicted based on cases presenting clinically [2]. This discrepancy is likely due to the high prevalence of individuals who are asymptomatic or experience only mild symptoms [4]. On the other hand, identifying asymptomatic individuals can impose a care burden and, more importantly, an emotional burden on patients and families [12,13], although specific studies on this pathology are lacking.

Consequently, the inclusion of 3-MCCD in NBS programs is currently controversial. In the United States, 3-MCCD is included in the Recommended Uniform Screening Panel (RUSP) endorsed by the Secretary’s Advisory Committee on Heritable Disorders in Newborns and Children. Conversely, countries such as Israel and Germany have excluded it due to its low benefit-to-harm ratio [4,14].

In Italy, a nationwide NBS program for inborn errors of metabolism is mandated by law. 3-MCCD is listed as a secondary condition to be considered in the differential diagnosis as it shares the C5OH marker with certain primary panel diseases including 3-hydroxymethyl-3-methylglutaryl-CoA lyase deficiency, β-ketothiolase deficiency and multiple carboxylase deficiency [15].

This study reports on over a decade of screening experience in north-east Italy and the long-term follow-up of positive cases. Our focus is on the overall impact of screening, investigating not only the clinical aspects, but also the psychological impact on parents. These insights aim to inform policy and healthcare decisions regarding the inclusion of 3-MCCD in newborn screening programs.

## 2. Materials and Methods

### 2.1. Newborn Screening

Between 1 January 2014 and 30 September 2025, 336,668 newborns were screened at the Regional Centre for Expanded Newborn Screening at Padua University Hospital. Following the NBS protocol, DBS samples were collected via heel prick and spotted on filter paper between 48 and 72 h after birth. A second sample was required for premature infants (those born before 34 weeks of gestation and/or weighing less than 2000 g) and sick newborns (those receiving transfusion or parenteral nutrition).

Samples were pre-processed according to the operating instructions for the non-derivatized MS/MS kit (NeoBase™, between 2014 and 2018; NeoBase™2, between 2019 and 2025) (PerkinElmer, Waltham, MA, USA). Quantitative analysis of acylcarnitines was performed using tandem mass spectrometry (MS/MS) screening systems (Acquity TQD, Waters, Milford, MA, USA and Qsight 210/225 MD, PerkinElmer, Waltham, MA, USA), following the manufacturers’ protocols.

C5OH was used as the primary biochemical marker for 3-MCCD screening, with a cut-off value ≥ 0.8 μmol/L (99.99th percentile).

Newborns with elevated C5OH concentrations were recalled for repeat testing. If the second test was also positive, the neonate was referred to the Clinic Unit for confirmatory tests. If newborns had two successive results indicating an increased C5OH level, their mothers’ blood C5OH levels were tested to exclude the transplacental passage of metabolites in cases of unrecognized maternal 3-MCCD.

### 2.2. Confirmatory Testing

Confirmatory testing included a DBS acylcarnitine profile using MS/MS, a spectrophotometric assay to measure plasma free carnitine levels [16], and a urinary organic acid analysis using gas chromatography/mass spectrometry (GC-MS) [17] to test for 3-HIVA and 3-MCG metabolites, as well as to exclude other metabolic disorders that present with elevated C5OH levels. Mutation analysis of the *MCCC1* and *MCCC2* genes was performed to confirm the diagnosis.

### 2.3. Management and Follow-Up

The treatment strategy focused on preventing fasting/metabolic stress during intercurrent illnesses. An emergency protocol letter was provided during the first visit and updated during follow-up. Close attention was paid to maintaining adequate caloric intake, with intravenous glucose administration when oral intake was impaired by illness.

Follow-up assessments were conducted every 12 months and included growth and development evaluations. Biochemical tests, including a complete blood count, blood gas evaluation, DBS C5OH, plasma free carnitine and urinary organic acids, were monitored. Oral carnitine supplementation was only administered if plasma levels were deficient. A psychologist specializing in inherited metabolic diseases (IMDs) was involved in communicating the diagnosis and subsequent follow-ups.

### 2.4. Psychological Assessment

We developed a qualitative survey consisting of six items designed to assess the following key areas: the initial emotional impact of screening; the clarity and accessibility of information provided by healthcare professionals; the perceived emotional support; the long-term parental emotional experiences; and the practical impact of the diagnosis on family life. One final item explored parents’ overall experience, focusing on their personal perceptions and the public utility they perceived in newborn screening (Table 1).

The questionnaire was constructed based on existing literature concerning the psychosocial impact of IMDs [18,19,20] with items adapted and refined to specifically address screening for and the psychosocial impact of 3-MCCD.

The survey was administered to parents during the last follow-up visit. Parents were asked to rate each item using a 10-point Likert scale (1 = not at all; 10 = very much). They were also invited to provide additional comments on their emotional experience, enabling a qualitative exploration of their perceptions and offering deeper insight into their psychological well-being.

### 2.5. Statistical Analysis

Continuous variables were presented as mean (standard deviation SD) or median (interquartile range). A Student’s *t*-test or a Mann–Whitney U test was used to compare groups of variables that were normally or non-normally distributed, respectively. All statistical analyses were conducted using GraphPad Prism version 5.00 (GraphPad Software, San Diego, CA, USA). A *p*-value of ≤0.05 was considered statistically significant.

### 2.6. Ethical Aspects

All caregivers provided informed consent to participate in the study, which was conducted in accordance with the principles of the Declaration of Helsinki. Approval for the study was granted by the Research Ethics Committee of the University Hospital of Padua (n. 62302, 16 September 2025).

## 3. Results

### 3.1. Newborn Screening and Confirmatory Testing

Between 1 January 2014 and 30 September 2025, we screened 336,668 newborns. Of these, 62 (43 males and 19 females) had elevated C5OH values ≥0.80 μmol/L cut-off point (mean 1.38 μmol/L, SD 1.12, range 0.80–5.59).

Of these, 34 cases normalized at the second DBS test. The remaining 28 cases were referred to the Clinic Unit.

At confirmatory testing, the C5OH values remained elevated in all patients (mean 1.89 μmol/L, SD 1.37 μmol/L, range 0.80–5.73 μmol/L).

Concurrent testing of the mothers identified nine cases (including two pairs of siblings) of maternal 3-MCCD. At birth, these newborns had mean C5OH values of 2.23 μmol/L (SD 1.37, range 0.85–4.48). At confirmatory testing, all had negative urinary organic acids (see Table A1), with progressive normalization of C5OH values during the first months of life.

In 18 other cases where biochemical abnormalities were confirmed and no maternal cause was identified, confirmatory tests were completed by genetic assays (see Table 2), except for one Moldovan newborn who returned to their country of origin before the tests could be carried out.

Eight patients (P1–P8), including two siblings, were found to carry two variants in either the *MCCC1* (*n* = 5) or *MCCC2* (*n* = 3) genes. At birth, their mean DBS C5OH level was 2.7 μmol/L (SD 2.17, range 0.85–6.59). Plasma free carnitine was reduced in five of the eight neonates. Urinary organic acids were positive for 3-MCG and 3-HIVA in six patients (five carrying *MCCC1* and one carrying *MCCC2* variants), positive only for 3-MCG in one *MCCC2* patient and negative in one *MCCC2* patient.

Seven patients were carriers of a heterozygous variant of the *MCCC1* gene (mean C5OH 0.96 μmol/L, SD 0.13, range 0.80–1.18; normal plasma free carnitine and negative urinary organic acids). Six patients were discharged (E1–E6), while one Italian female patient (P9) is followed due to a variant associated with a dominant negative effect (*MCCC1* gene, p.Arg385Ser).

Three patients (N1–N3) had negative genetic assays. Biochemical tests showed a mean C5OH at birth of 1.30 μmol/L (SD: 0.19, range: 1.08–1.44), with normal plasma free carnitine and negative urinary organic acids. Their C5OH values progressively decreased, and they were discharged.

Analysis of C5OH at birth showed no significant difference between true positives and maternal cases (*p* = 0.6). Combined, true positives and maternal cases were significantly different from false positive cases (*p* = 0.02), but there was overlap. No significant differences were found in DBS carnitine (C0) values, including between true positives (mean 20.06 μmol/L, SD 8.91) and maternal cases (mean 14.49 μmol/L, SD 8.70). However, there was a lower trend in maternal cases.

### 3.2. Follow-Up

All nine true positive patients (P1–P9) were periodically monitored. At their last visit, they had a mean age of 4.22 years (SD 3.73, range 1–10 years) (see Table 3).

Five patients started carnitine supplementation between birth and three years of age due to reduced levels of plasma free carnitine (mean 8.85 μmol/L, SD 3.17, range 6.25–14).

One patient experienced metabolic decompensation involving acidosis, hypoglycemia and ketosis (pH 7.25, PCO_2_ 30 mmHg, HCO_3_^−^ 14 mmol/L, BE −10 mmol/L, lactate 2 mmol/L, ketones 3.6 mmol/L) during an intercurrent infection at one year of age. Metabolic investigations revealed low free carnitine levels (6.6 μmol/L despite receiving 100 mg/kg of carnitine supplementation), C5OH levels of 29.41 μmol/L and urinary organic acids indicative of severe ketosis, dicarboxylic aciduria and elevated 3-MCG and 3-HIVA levels. The patient was treated with a glucose infusion and a transient increase in the dosage of carnitine (200 mg/kg/day), resulting in a rapid improvement.

Another patient showed psychomotor delay in the early years (at 2 years of age, Bayley III cognitive score of 85, speech score of 79 and motor function score of 73). This patient also required carnitine supplementation due to low carnitine levels (8.4 μmol/L) from the age of two. It is possible that the carnitine deficiency contributed at least in part to the motor delay. Intelligence quotient (IQ) later normalized (5.5 years, WPPSI-III IQ 92; 7 years, WISC-IV IQ 95).

Finally, a patient presented with poor growth (2 years: weight 8700 g, <3° pc according with CDC, height 79.5 cm, 10° pc according with CDC, head circumference 45 cm, 50–75° pc according with CDC) and carnitine deficiency (6.3 μmol/L). However, she also suffered from neurofibromatosis type I, which is another potential cause of poor growth.

Notably, all reported cases presented with clinical manifestations and carnitine deficiency in the first years of life.

At the last follow-up, the mean C5OH value was 9.79 μmol/L (SD 13.80, range 1.07–45.53). All patients tested positive for urinary 3-MCG and 3-HIVA. Carnitine levels tended to be low (mean 28.68 μmol/L, SD 11.85, range 14–49.7), but only one patient (P4) had severe deficiency (14 μmol/L) and had started supplementation by that point.

Patients undergoing carnitine therapy exhibited higher C5OH levels (15.73 μmol/L, SD 16.73, range 5.89–45.53) than those not undergoing therapy (2.37 μmol/L, SD 1.44, range 1.07–4.4), though this difference was not statistically significant, probably due to the small sample size. There was a difference between the two groups before starting supplementation (patients who required therapy due to low free carnitine levels had a mean C5OH level of 8.09 μmol/L, SD 2.79, range 5.72–12.55, *p* = 0.008).

We analyzed a possible correlation between C5OH values at NBS and the development of symptoms. Patient P1 had the highest C5OH level (6.59 μmol/L) in the NBS DBS among our patients and the lowest plasma free carnitine level (6.25 μmol/L). The other two patients with higher C5OH values at birth (5.32 and 2.89 μmol/L, respectively) presented with delayed motor development and poor growth, respectively.

Furthermore, patients who later required carnitine therapy had higher C5OH levels at birth (mean C5OH: 3.72 μmol/L, SD: 2.18, range: 1.21–6.59) compared to the untreated group (C5OH: 0.94 μmol/L, SD: 0.16, range: 0.80–1.16, *p* = 0.04).

### 3.3. Psychological Assessment

A total of 14 out of 16 parents of 3MCCD patients (P1-P9, including siblings P3 and P8) participated in the survey. Eight of the participants were fathers and six were mothers; two additional mothers did not participate due to language barriers. The results are summarized in Table 4 and detailed in Table A2.

Question 1: How anxious did you feel after receiving the result of the newborn screening?

All parents reported high levels of anxiety following a positive result, with mothers showing significantly greater concern than fathers (median score for mothers = 10 vs. fathers = 7.5, *p* = 0.02).

Parents consistently described a strong sense of anxiety, primarily in relation to receiving the initial phone call about the positive result for the metabolic condition, and the confusion that followed. One mother characterized this moment as *‘a real blow’*.

Question 2: How easy did you find it to understand the information you were given about the disease?

Parents reported some difficulties in understanding the information they received (median score of 5.5 for mothers and 5 for fathers).

Many parents distinguished between two key moments: (a) the initial phone call from the neonatologist informing them of the positive screening result and the need to attend a specialized center for confirmatory testing, and (b) the subsequent visit to the IMD specialist center.

The first phone call was perceived as the most confusing and difficult. Five parents (35.7%) explicitly reported problems such as a lack of medical knowledge, the use of overly technical language and insufficient essential information. The confusion following this initial contact was mentioned repeatedly. One father explained: ‘They were not clear. They didn’t give us the information we needed. Without explanations, you start imagining the worst, especially because you are a new parent.’

By contrast, parents reported clearer and more comprehensible communication when detailed information about the disease was provided at the IMD center. They attributed this improvement to the calmer environment, the availability of a specialized medical staff, and the opportunity to ask questions. One father emphasized that building a trusting relationship with physicians was crucial for understanding the disease.

Question 3: How much emotional support did you feel you received from healthcare professionals?

Both mothers (median = 8) and fathers (median = 8) perceived that they received adequate emotional support from healthcare professionals.

Indeed, the majority of parents (10 out of 14, or 71.4%) reported receiving adequate emotional support from all professionals in the specialized IMD team.

Question 4: How often do you worry about your child’s future health due to their diagnosis?

Mothers and fathers reported some concerns about their child’s future health (median: mothers = 5.5; fathers = 4).

Several indicated that they were much more worried immediately after receiving the diagnosis, particularly due to a poor knowledge about the disease (“At first, we didn’t know what the condition was”), but that this concern had diminished over time as they had learned more about its characteristics and severity (“Initially, we were very worried, but now we are calmer”). Many felt reassured by the availability of an emergency treatment and regular follow-ups (‘Now I feel reassured because we are monitored, and the doctors tell us what to do’). One parent expressed the ambivalence of living with the condition: ‘They told us it is not a disease that causes problems for the child, but it is still a disease.’

Q5: How much did the diagnosis initially/currently affect your or your family’s daily life?

Parents, particularly mothers, described the diagnosis having a strong impact on their daily lives immediately after receiving the result (median score for mothers = 7.5, for fathers = 4.5). The current impact of the diagnosis was reported to be minimal for both caregivers (median score for mothers = 3, for fathers = 1).

Parents reported a strong initial impact on daily life, describing feelings of worry and disorientation when returning to everyday routines after learning about their child’s condition. Mothers, in particular, highlighted how these difficulties were exacerbated by the challenges of the postpartum period.

Over time, however, parents said that the current impact of the disease on daily life was minimal. They reported having gained greater knowledge of the condition and answers to their initial questions. They also reported having developed practical skills to manage their child’s needs, which reduced the impact on family life.

Q6: How useful do you think newborn screening for this disease has been in your experience?

All parents considered newborn screening for the condition to be highly useful (median: mothers = 10, fathers = 8). Despite this positive overall perception, however, mothers rated the usefulness of the test significantly higher than fathers did (*p* = 0.02).

They valued the early diagnosis, as it enabled them to care for their child proactively and prepare for the future. As one father stated, ‘Knowing about the problem helps you to manage the disease itself; it is a warning signal that helps you to prevent possible difficulties.’

## 4. Discussion

We present the outcomes of our ten-year experience managing and following up with patients identified with 3-MCCD through newborn screening, addressing both clinical and psychological aspects.

Of the 336,668 newborns screened, 28 were referred for a clinical evaluation, and nine were confirmed as affected (predictive positive value PPV 15%). 18 were found to be unaffected, including nine maternal cases, six heterozygotes, and three who tested negative on molecular examination. One patient was lost to follow-up before molecular testing. The biochemical parameters at newborn screening overlapped among the various groups, and only the molecular examination was conclusive.

Of the nine affected individuals, one was homozygous, seven were compound heterozygotes, and one (P9) was heterozygous for the p.Arg385Ser variant, which is a dominant-negative allele that leads to biochemical abnormalities and clinical symptoms in heterozygous individuals [21,22].

Since the first visit, the parents were instructed in the management of the patient during metabolic stress/intercurrent infections, and the patients underwent annual clinical and biochemical follow-up.

During the follow-up, one patient (P1) experienced acidosis, ketosis, hypoglycemia, and carnitine deficiency at one year of age, during an infectious episode accompanied by vomiting and inability to follow the oral emergency protocol. The patient was hospitalized and promptly recovered through the administration of an intravenous glucose solution to provide high caloric intake and carnitine supplementation, thus avoiding progression to severe decompensation. This case certainly benefited from newborn screening. Furthermore, it is unclear whether other patients decompensated without alert protocols during intercurrent illnesses. The literature reports that many newborns diagnosed through NBS remain asymptomatic as long as they adhere to the appropriate treatment regimen. However, patients who are lost to follow-up or who are noncompliant tend to succumb to ketoacidosis during acute stress, such as infection or trauma [4,23,24]. These individuals are at risk for mortality or long-term disability during episodes of metabolic decompensation [4].

Furthermore, five out of nine patients developed carnitine deficiency, requiring oral supplementation, during the first years of life. This is due to increased renal excretion of C5OH, which is a natural way of excreting toxic intermediary metabolites [4]. It is known that secondary carnitine deficiency can cause a disorder of fatty acid metabolism, leading to a series of clinical symptoms [25]. Dilated cardiomyopathy associated with carnitine deficiency has been reported in infants with 3-MCCD [26], which highlights the importance of monitoring and supplementation when necessary. This demonstrates a second benefit affecting 55% of patients in our population. Of the five patients, P1 had severe carnitine deficiency since birth. Other two patients with carnitine deficiency had growth retardation and transient psychomotor delay, respectively. It is possible that the carnitine defect was a contributing factor, making supplementation particularly important.

Our data align with those of Grunert et al., who identified 36 subjects with carnitine deficiency through NBS [27]. Of these subjects, 31% developed clinical symptoms, including various neurological symptoms, and 14% experienced at least one episode of acute metabolic decompensation. Sixty percent had decreased free carnitine levels.

To maintain these benefits but reduce the medicalization of asymptomatic patients, it should be important to identify predictive factors for clinical manifestations. Our patient who presented with decompensation had the highest C5OH value on NBS; patients who developed carnitine deficiency also had higher C5OH values at birth. However, the data from literature are conflicting [27,28,29], and our finding requires confirmation in larger populations.

To summarize the clinical point of view, our study demonstrates the relevance of early detection through NBS to establish treatment for metabolic stress situations, such as recurrent infections, to prevent metabolic decompensation and carnitine deficiency.

However, this must be balanced against the potential harm that NBS could cause due to unwanted stigmatization or unnecessary interventions. For these reasons, some authors and decision-making authorities believe that the disadvantages outweigh the advantages [4,14]. Although the psychological aspects of positive newborn screening have been extensively studied [12,13], specific data on this particular condition is unavailable.

In our population, we investigated the psychological aspects through a qualitative survey. All parents reported experiencing high levels of anxiety immediately after receiving the NBS results. Mothers reported greater anxiety than fathers, likely due to the postpartum period and its associated physical, hormonal, and psychological changes. Most parents indicated that receiving a positive screening result over the phone from a neonatologist was the most stressful moment, “a real blow.” Parents reported a lack of essential information and clarity about their newborn’s medical condition, as well as the use of highly technical language. This lack of information led many parents to feel confused and insecure, sometimes “imagining the worst” about their child’s health.

These findings underscore the importance of reflecting on the content and delivery of the initial phone call to parents. A review by Tluczek et al. showed that many parents are uninformed about NBS. Specifically, mothers often did not receive prenatal education and were unaware of how they would receive NBS results or the implications of abnormal results. This led to uncertainty about the appropriate actions to take after the recall [30]. Similarly, DeLuca et al. reported that the initial response to a call announcing a positive NBS result was often characterized by shock, difficulty comprehending the information provided, and a sense of insecurity due to providers’ insufficient knowledge of the disorders [31]. Specific training and greater awareness among healthcare professionals about communicating initial positive results is essential. Additionally, providing information through videos and brochures about NBS, especially during the last trimester of pregnancy, improves understanding and fosters more positive parental attitudes [32].

Moreover, in our population parents reported that their child’s 3-MCCD diagnosis impacted their individual and family life in the period immediately following the diagnosis, though this impact decreased over time and was minimal by the last follow-up visit. The results suggest that the disease’s influence extends beyond the initial shock and affects family life, particularly during the early stages of a child’s development. However, parents reported that they were able to cope with these difficulties due to their increased knowledge of the condition, answers to their initial questions, and acquisition of practical skills to manage their child’s needs. This is also because parents reported receiving adequate emotional support from the multidisciplinary specialist team. This adaptation process has also been described in other metabolic disorders. For example, Siddiq et al. found that, while some parents experienced difficulty adapting and stress during the process, most were able to integrate the disease into daily family life by using proactive coping strategies [19]. Our previous study of the psychological aspects of asymptomatic type I Gaucher disease diagnosed through NBS also demonstrated an adaptation process during the first years of life [33].

These findings underscore the importance of comprehensive care from a multidisciplinary team that supports parents, especially during the initial phases following recall and diagnosis. This support provides parents with a listening ear and psychological assistance, helping them develop adaptive coping strategies and facilitating the family’s adjustment to the chronic condition.

Finally, all parents strongly believed in the usefulness of newborn screening for their children. Many explained that NBS is essential for an early diagnosis, which allows them to care for their child and avoid a worse outcome. This is according to other studies on IMD, that have shown that most parents support routine NBS and value receiving information about their child’s health or their own carrier status [30,31].

The limitations of our study include the small sample size and the fact that it is a cross-sectional study regarding psychological aspects. Further studies on larger populations conducted prospectively are needed to confirm our data. Moreover, similar studies could be extended to other disorders included in NBS programs, such as Short-Chain Acyl-CoA Dehydrogenase Deficiency (SCADD).

## 5. Conclusions

In conclusion, newborn screening for 3-MCCD offers advantages in preventing acute decompensation and providing carnitine supplementation when necessary. Although there is an initial psychological burden on the family, it reduces over time and has only a minimal impact on daily life during follow-up. Some aspects can be improved, particularly early identification of patients with high risk of metabolic decompensation and improvement of initial communication. Despite these challenges, we believe that newborn screening is useful for managing these patients and that the clinical benefits outweigh the psychological burden.

## Figures and Tables

**Figure 1 IJNS-11-00115-f001:**
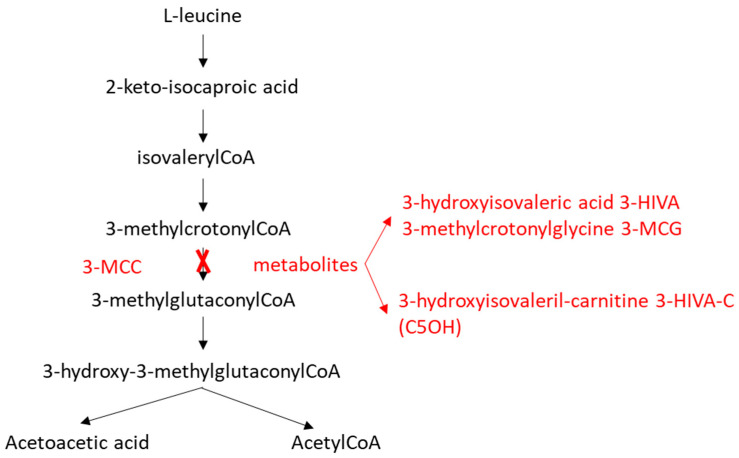
The 3-MCC-catalyzed reaction in the leucine catabolic pathway.

**Table 1 IJNS-11-00115-t001:** Ad hoc qualitative survey to test the psychological impact of NBS on parents of children with 3-MCCD.

Item	Question (Rate from 1 to 10, Where 1 = Not at All, 10 = Very Much)
Q1	How anxious did you feel after receiving the result of the newborn screening?
Q2	How easy did you find it to understand the information you were given about the disease?
Q3	How much emotional support did you feel you received from healthcare professionals?
Q4	How often do you worry about your child’s future health due to the diagnosis?
Q5a	How much did the diagnosis initially affected your or your family daily life?
Q5b	How much does the diagnosis currently affect your or your family daily life?
Q6	How useful do you think newborn screening for this disease has been in your experience?

Additional Comments: Please provide any additional thoughts or experiences regarding the emotional impact of the screening and diagnosis process.

**Table 2 IJNS-11-00115-t002:** Clinical, Biochemical, and Molecular Features of Neonates Positive for 3-MCCD.

Case	Sex	Ethnic Origin	NBS		Biochemical Phenotype			Genotype					Diagnosis
			DBS C5OH (Cut off ≥0.80 µmol/L)	DBS C0 (nv > 11 µmol/L)	DBS C5OH (0.08–0.50 µmol/L)	Plasma Free Carnitine (29–42 µmol/L)	Urinary Organic Acids	Affected Gene	Nucleotide Change 1	Nucleotide Change 2	Amino Acid Change 1	Amino Acid Change 2	
P1	M	Italy	6.59	18.5	5.73	6.25	3-MCG, 3-HIVA	*MCCC1*	c.189G>C	c.1331G>A	p.Arg63Ser	p.Arg444His	3-MCCD
P2	F	Italy	5.32	28.9	3.18	25.6	3-MCG	*MCCC2*	c.358A>T	c.436T>C	p.Ile120Phe	p.Tyr146His	3-MCCD
P3	M	Italy	0.96	8.5	1.28	13.2	3-MCG, 3-HIVA	*MCCC1*	c.230C>T	c.1526del	p.Ala77Val	p.Cys509Serfs*14	3-MCCD
P4	M	Bangladesh	2.6	33.4	2.91	32.8	negative	*MCCC2*	c.387T>C	c.903+5G>A	p.Ile133Thr	p.?	3-MCCD
P5	M	Italy	1.21	13.2	1.98	31.1	3-MCG, 3-HIVA	*MCCC1*	c.1194A>C	c.1594+1G>A	p.Glu383Asp	p.?	3-MCCD
P6	M	Morocco	0.85	18.9	1.16	49.6	3-MCG, 3-HIVA	*MCCC1*	c.1008G>C	c.1008G>C	p.Met336Ile	p.Met336Ile	3-MCCD
P7	F	Italy	2.89	30.9	5.33	22.7	3-MCG, 3-HIVA	*MCCC2*	c.295G>C	c.724G>T	p.Glu99Gln	p.Ala242Ser	3-MCCD
P8	M	Italy	1.16	12.4	1.30	15.3	3-MCG, 3-HIVA	*MCCC1*	c.230C>T	c.1526del	p.Ala77Val	p.Cys509Serfs*14	3-MCCD
P9	F	Italy	0.80	15.8	0.81	24.9	negative	*MCCC1*	c.1155A>C	/	p.Arg385Ser	/	3-MCCD (Dominant negative allele)
E1	M	Kosovo	0.92	14.1	0.81	27.1	negative	*MCCC1*	c.725_727delATG	/	p.Asp242del	/	Healthy carrier
E2	F	East Asia	1.01	18	1.21	33.2	negative	*MCCC1*	c.863A>G	/	p.Glu288Gly	/	Healthy carrier
E3	M	Moldova	1.05	20.2	1.02	35.4	negative	*MCCC1*	c.1155A>C	/	p.Arg385Ser	/	Healthy carrier
E4	M	Italy	0.92	32	0.80	34.9	negative	*MCCC1*	c.414delT	/	p.Phe138Leufs*13	/	Healthy carrier
E5	F	Albania	0.87	18.7	0.84	31.6	negative	*MCCC1*	c.288T>A	/	p.Tyr96*	/	Healthy carrier
E6	M	Albania	1.18	17	1.45	29.4	negative	*MCCC1*	c.1399A>T	/	p.Ile467Phe	/	Healthy carrier
N1	F	Kosovo	1.44	50.5	1.06	34.9	negative	Negative					
N2	M	Italy	1.37	19.3	1.06	33.2	negative	Negative					
N3	F	Moldova	1.08	34.6	0.9	40.9	negative	Negative					

**Table 3 IJNS-11-00115-t003:** Clinical and Biochemical Follow-Up of Confirmed 3-MCCD Patients.

	Clinical Phenotype/Outcome		Last Follow Up						Biochemical Parameters Before Carnitine Supplementation	
	Metabolic Decompensation, Age	Growth and Development	Current Age (y)	DBS C5OH (0.08–0.50 µmol/L)	Plasma Free Carnitine (29–42 µmol/L)	Urinary Organic Acids	Diet	Medication	Age at Start Therapy	Plasma Free Carnitine (29–42 µmol/L)
P1	1 yrs	Normal	10	45.53	22.7	3MCG, 3HIVA	free	Carnitine 100 mg/kg	At birth	6.25
P2	No	Developmental delay (transient)	10	9.70	23.3	3MCG, 3HIVA	free	Carnitine 100 mg/kg	2 yrs	8.4
P3	No	No	7	4.4	29.9	3HIVA, 3MCG	free	no	/	/
P4	No	No	3	7.98	14	3HIVA, 3MCG	free	no	3 yrs	/
P5	No	No	2	5.89	44.4	3MCG, 3HIVA	free	Carnitine 100 mg/kg	3 mos	9.3
P6	No	No	2	1.71	20.7	3MCG, 3HIVA	free	no	/	/
P7 *	No	Growth retardation	2	9.56	20.3	3MCG, 3HIVA	free	Carnitine 100 mg/kg	1 yr	6.3
P8	No	No	1	2.29	33.1	3MCG, 3HIVA	free	no	/	/
P9	No	No	1	1.07	497	3MCG, 3HIVA	free	no	/	/

* comorbidity with neurofibromatosis type I.

**Table 4 IJNS-11-00115-t004:** The psychological assessment results of the parents of 3-MCCD patients identified through newborn screening.

	Fathers (*n* = 8)	Mothers (*n* = 6)	
	Median	Median	*p*-Value
Question 1	7.5	10	0.02 *
Question 2	5.0	5.5	0.89
Question 3	8.0	8.0	0.79
Question 4	4.0	5.5	0.22
Question 5a	4.5	7.5	0.20
Question 5b	1.0	3.0	0.52
Question 6	8.0	10	0.02 *

* statistically significant (*p* < 0.05).

## Data Availability

The data supporting the findings of this study are available from the corresponding author upon request.

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
