# Peer review of "Psychological Impact of Newborn Screening for 3-Methylcrotonyl-CoA Carboxylase Deficiency: The Parental Experience"

_2409-515X, 2025, doi:10.3390/ijns11040115_

Round 1

Reviewer 1 Report

Comments and Suggestions for Authors

The authors have studied the psychological impact of a positive neonatal screening result for 3-MCCD on the parents of the referred newborn. They also provide a very useful and detailed overview of the referrals that have happened in screening more than 336,000 babies and their follow up.

The work is well written and of interest especially to other screening programs that have implemented screening for this disorder, because a large fraction of individuals with this disorder are believed to remain asymptomatic.

I have only a few small suggestions for the manuscript:

  • in table A1, what is the meaning of the abbrevation "nv"? (in the columns with C0 data)
  • in table A1 or it's legend, please state what the concentration range in the column headers means. (is this the normal concentration range?)
  • The cutoff for C5OH must be 'greater than or equal to' (≥), but this is only clear because one of the positive results was exactly 0.8. It would be preferable to state this explicitly wherever the cutoff is mentioned (page 3 line 102 /  page 5 line 160-161 / table A1)

Author Response

The authors have studied the psychological impact of a positive neonatal screening result for 3-MCCD on the parents of the referred newborn. They also provide a very useful and detailed overview of the referrals that have happened in screening more than 336,000 babies and their follow up.

The work is well written and of interest especially to other screening programs that have implemented screening for this disorder, because a large fraction of individuals with this disorder are believed to remain asymptomatic.

I have only a few small suggestions for the manuscript:

in table A1, what is the meaning of the abbrevation "nv"? (in the columns with C0 data)

in table A1 or it's legend, please state what the concentration range in the column headers means. (is this the normal concentration range?)

R: Thank you for your valuable comments. We appreciate the opportunity to clarify:

In Table A1, we have added a note in the legend stating: 'nv: normal value' to explain the abbreviation.

Regarding the concentration ranges, we will add a note in the table legend stating: 'The concentration ranges shown in the column headers represent the normal reference ranges for each parameter.'

The cutoff for C5OH must be 'greater than or equal to' (≥), but this is only clear because one of the positive results was exactly 0.8. It would be preferable to state this explicitly wherever the cutoff is mentioned (page 3 line 102 /  page 5 line 160-161 / table A1)

R: Thank you for your observation regarding the C5OH cutoff. As per your suggestion, we have made the following changes: We have replaced the cutoff description for C5OH with the 'greater than or equal to' (≥) symbol wherever it is mentioned in the manuscript. Specifically, we have made this change: On page 3, line 102, On page 5, lines 160-161, In Table A1.

Reviewer 2 Report

Comments and Suggestions for Authors The basic question addressed in the reviewed work is the suitability of 3-methyl crotonyl CoA deficiency (3-MCCD) for nationwide neonatal screening (NS). This question is relevant and fundamental, because 3-MCCD only marginally meets the Wilson-Jungner (W-J) criteria for inclusion in regular NS. The incidence of 3-MCCD is low and most cases are asymptomatic. On the other hand, positive cases are at risk during intercurrent illnesses. The presented work documents, in the cases of monitored patients with 3-MCCD, their critical threat during intercurrent illnesses with the need for inclusion in NS and adequate information not only for the child's parents, but also for the relevant primary care physician - with long-term follow-up and possible carnitine supplementation. The conclusions of the work are in accordance with the analysis of the set and document the fact that parents of affected children appreciate the importance of NS and the initial stress from the information does not affect their opinion. The issue of psychological stress of parents due to information about a positive screening test has been addressed in several works (e.g. Muir Starmer e.al.) and my experience as a long-time pediatrician is that the solution to this issue lies with primary care pediatricians and adequate information for parents. The initial concern of parents – who sometimes drive their child in a car without seat belts – cannot outweigh the need for NS for diseases that may affect health throughout life. The authors' arguments are understandable and unambiguous – neonatal screening 3-MCCD belongs to the spectrum of IMD with adequate information for parents and healthcare professionals. The work provides a clear answer and also space for professional discussion.

Similar situation is in SCADD newborn screening

Author Response

The basic question addressed in the reviewed work is the suitability of 3-methyl crotonyl CoA deficiency (3-MCCD) for nationwide neonatal screening (NS). This question is relevant and fundamental, because 3-MCCD only marginally meets the Wilson-Jungner (W-J) criteria for inclusion in regular NS. The incidence of 3-MCCD is low and most cases are asymptomatic. On the other hand, positive cases are at risk during intercurrent illnesses. The presented work documents, in the cases of monitored patients with 3-MCCD, their critical threat during intercurrent illnesses with the need for inclusion in NS and adequate information not only for the child's parents, but also for the relevant primary care physician - with long-term follow-up and possible carnitine supplementation. The conclusions of the work are in accordance with the analysis of the set and document the fact that parents of affected children appreciate the importance of NS and the initial stress from the information does not affect their opinion. The issue of psychological stress of parents due to information about a positive screening test has been addressed in several works (e.g. Muir Starmer e.al.) and my experience as a long-time pediatrician is that the solution to this issue lies with primary care pediatricians and adequate information for parents. The initial concern of parents – who sometimes drive their child in a car without seat belts – cannot outweigh the need for NS for diseases that may affect health throughout life. The authors' arguments are understandable and unambiguous – neonatal screening 3-MCCD belongs to the spectrum of IMD with adequate information for parents and healthcare professionals. The work provides a clear answer and also space for professional discussion. Similar situation is in SCADD newborn screening.

R: Thank you for your comprehensive and insightful review. We appreciate your expert perspective on the importance of neonatal screening for 3-MCCD, despite its challenges against the Wilson-Jungner criteria.

We agree with your observations on parental stress management and the crucial role of adequate information for both parents and healthcare professionals.

Following your valuable comparison to SCADD, we have added the following to our discussion: 'Similar studies could be extended to other disorders included in neonatal screening programs, such as Short-Chain Acyl-CoA Dehydrogenase Deficiency (SCADD).'